

# Uncertainties originating from GCM downscaling and bias correction with application to the MIS-11c Greenland Ice Sheet

Brian R. Crow[1], Lev Tarasov[2], Michael Schulz[1], Matthias Prange[1]

[1]MARUM (Center for Marine Environmental Sciences) and Faculty of Geosciences, University of Bremen, Bremen, 28359, Germany

[2]Department of Physics and Physical Oceanography, Memorial University of Newfoundland and Labrador, St. John's, A1B 3X7, Canada

*Correspondence to*: Brian R. Crow (bcrow@marum.de)





**Abstract.** The Marine Isotope Stage 11c (MIS-11c) interglacial is an enigmatic period characterized by a long duration of
relatively weak insolation forcing, but is thought to have been coincident with a large global sea level rise of 6-13 m. The
configuration of the Greenland Ice Sheet during the MIS-11c interglacial highstand is therefore of great interest. Given the
limited data constraints, model-based analysis may be of use, but only if model uncertainties are adequately accounted for. A
particularly under-addressed issue in coupled climate and ice sheet modeling is the coupling of surface air temperatures to the
ice model. Many studies apply a uniform "lapse rate" accounting for the temperature differences at different altitudes over the
ice surface, but this uniformity neglects both regional and seasonal differences in near-surface temperature changes. Herein
we provide the first such analysis for MIS-11c Greenland that addresses these uncertainties by comparing 1-way coupled
CESM and ice sheet model results from several different downscaling methodologies.

In our study, a spatially- and temporally-varying temperature downscaling method produced the greatest success rate in
matching limited paleodata constraints, and suggests a peak ice volume loss from Greenland during MIS-11c of near 50%
compared to present day (~3.9 m contribution to sea level rise). This result is on the lower bound of existing data- and model-
based studies, partly as a consequence of the applied one-way coupling methodology which neglects some feedbacks.
Additional uncertainties are examined by comparing two different present-day regional climate analyses for bias correction of
temperatures and precipitation, a spread of initialization states and times, and different spatial configurations of precipitation
bias corrections. No other factor exhibited greater influence over the simulated Greenland ice sheet than the choice of
temperature downscaling scheme.

## 1 Introduction

Examining past interglacial climates offers the opportunity to conduct data-based tests of our understanding of ice-climate
dynamics and the modeling thereof. With present and near-future warming expected to further accelerate ice loss from the
Greenland and Antarctic ice sheets, maximizing our understanding of their behavior under past warm conditions is a necessity.
The Marine Isotope Stage 11c (MIS-11c) interglacial, spanning approximately 430,000 to 395,000 years ago (ka), presents a
particularly interesting test case for modelers given evidence of a robust sea-level highstand (i.e., large loss of land-ice mass)
despite relatively weak insolation forcing (Dutton et al., 2015; Tzedakis et al., 2022).

The relative contributions of the Greenland and Antarctic ice sheets to sea level rise during MIS-11c remain poorly constrained,
but recent modeling work has proposed plausible Greenland contributions in the range of 3.9-7.0 m (Robinson et al., 2017).
However, the relative importance of various forms of uncertainty are largely unaccounted for in many coupled ice-climate
studies, particularly with regards to bias corrections and temperature downscaling. In this study, we therefore illustrate the
dependence of a simulated MIS-11c Greenland ice sheet (GrIS) upon the key choices made with regards to the simulated
climate forcing and its coupling to an ice model.



One approach for examining the ice-climate interactions through glacial-interglacial cycles is the two-way interactive coupling of Earth-system models of intermediate complexity (EMICs) and ice-sheet models, enabling direct feedback of e.g., albedo, vegetation, land surface, and elevation changes on the climate forcing (e.g., Ganopolski and Calov, 2011; Goelzer et al., 2016; Robinson et al., 2017; Bahadory and Tarasov, 2018). Such coupled setups benefit from being computationally efficient, enabling long runs and often large ensembles of numerous simulations. However, many model components are highly simplified, and they can therefore only reproduce large-scale features of glacial-interglacial cycles. In recent years, more sophisticated but computationally expensive atmosphere-ocean general circulation models (AOGCMs) have also been increasingly used in two-way coupled setups (e.g., Ridley et al., 2005; Helsen et al., 2013; Sommers et al., 2021). Asynchronous acceleration techniques, in which the ice component is run for multiple years to multiple millennia before updated ice and climate states are exchanged via the coupler, are able to reduce the overall simulation time for such setups (e.g., Herrington and Poulsen, 2011; Helsen et al., 2013; Sommers et al., 2021). While future studies should ideally strive towards more fully-coupled simulations, AOGCM-based coupled simulations remain very computationally demanding at present, and this effectively precludes the possibility of conducting large ensemble simulations.

One common, computationally simpler alternative involves the one-way (offline) coupling of an AOGCM to an ice sheet model. Climate forcing is typically calculated using a steady-state present day or other prescribed ice sheet, limiting the direct feedbacks that the melt or growth of the ice sheet would actually have on the climate system (see e.g. Fyke et al., 2018 for a comprehensive overview). For any given lengthier period of interest, a series of several shorter-duration simulations at conditions representative of selected critical timesteps can be run, with the forcing then interpolated to be continuous between these slices (Stone et al., 2013). Such an approach can be useful for simulating conditions spanning full interglacials (e.g., Stone et al., 2013; Milker et al., 2013) or for comparing various interglacials to each other (Herold et al., 2012; Rachmayani et al., 2016; Rachmayani et al., 2017). This time-slice approach is what we have opted for in our study on the basis that it enables us to test a variety of coupling methodologies in a computationally efficient manner.

Regardless of chosen modeling approach, the relatively low-resolution surface temperatures simulated by a climate model must then be downscaled to the higher-resolution ice surface via a lapse rate, or rate of change in temperature with height. Typically this is a prescribed scalar value (e.g., Huybrechts, 1997, Viscaino et al., 2008) or a tunable parametric value (Stone et al., 2010), but neither of these options have a justifiable physical basis. Both methods fail to capture the considerable seasonality and regional variation of lapse rates that has been demonstrated by both in-situ measurements and model simulations of temperatures over glaciers and ice sheets (e.g., Gardner et al., 2009; Fausto et al., 2009; Erokhina et al., 2017). The high sensitivity of ice sheet marginal ablation zones to temperature changes (e.g., Stone et al., 2010), and the control the lapse rate exhibits on the strength of the temperature-elevation feedback, implies a strong need to correctly implement this in model simulations.





The MIS-11c interglacial specifically constitutes a particularly challenging target for a modeling study due to the relative lack of geological constraints on the extent of the ice sheets. Among the limited geological constraints on Greenland's extent dating back to MIS-11c are ice core samples near Summit and DYE-3. Chemical analysis of silty basal ice and bedrock beneath it

from the GISP2 core near Summit have suggested the possibility of some limited ice-free time over the past 2.7 million years (Schaefer et al., 2016), but it is likely that most or all of this time preceded the mid-Pleistocene transition (Bierman et al., 2014; Bierman et al., 2016; Yau et al., 2016). Previous model simulations have suggested that the GISP2/Summit region would be among the last places in Greenland to deglaciate even during exceptionally warm stretches (Fyke et al., 2014), suggesting that disappearance of ice at this location would be tantamount to the virtually complete loss of the GrIS. The basal ice at DYE-3,

however, has been dated only to the end of the MIS-11 interglacial, albeit with considerable uncertainties arising from dating techniques and poorly constrained ice advection (Yau et al., 2016). Thus, directly derived constraints for the minimum extent of the GrIS during MIS-11c include (1) the preservation of ice at Summit and (2) the disappearance of ice at DYE-3.

Additional indirect evidence of GrIS deglaciation in MIS-11c originates from marine sediment cores from a handful of

90 locations off southern Greenland. Spruce pollen found in these samples, considered to be of local origin, indicates the emergence of boreal coniferous forest across at least the lower elevations of southern Greenland sometime around 400 ka. This is roughly during the later stages of MIS-11c and suggestive of considerable retreat of the GrIS ice margin compared to present (Willerslev et al., 2007; de Vernal and Hilliare-Marcel, 2008). Cessation of ice-rafted debris (IRD) deposition along the southern margin for several thousand years during MIS-11c is unprecedented compared to other late Pleistocene interglacials,

and indicative of the disappearance of most or all marine-terminating ice in southern Greenland (Reyes et al., 2014). Collectively, this evidence suggests a drastic reduction in the extent of the GrIS, but with rather poor constraints on the magnitude, spatial extent, or duration of retreat.

In this study, we present a number of one-way coupled ensemble simulations of the Greenland ice sheet's evolution throughout

its substantial melt event during the MIS-11c interglacial. Using constraints provided by reconstructions, we determine a likely range for the GrIS contribution to sea-level change during MIS-11c. We examine the sensitivity of the simulated GrIS to a range of options, including those that are more observationally and physically justifiable than what has generally been used to date. In particular, we demonstrate that commonly-used scalar lapse rates for temperature downscaling perform poorly against our data-based constraints and produce the least GrIS melt in MIS-11c of any tested scheme. Our downscaling techniques,

bias-correction schemes, initialization states, and chosen models are all detailed in the following section.



## 2 Methodology

The present study is centered on the one-way coupling of the climate forcing developed in the Community Earth System Model (CESM) v.1.2.2 with the ice dynamics of the Glacial Systems Model (GSM). A one-way coupling methodology (i.e., CESM forcing provided to GSM with no coupling back to CESM) was selected for computational efficiency reasons; namely, iterative CESM topographic corrections between time slices was judged too impractical to implement, some feedbacks would still be lacking compared to full 2-way coupling, and running a large ensemble of simulations would not be feasible. Relevant descriptions of the two models and the selection and processing of key input variables follow. The purpose here is to overview the various techniques we utilized as they pertain to the treatment of climate forcing and their coupling to the ice model.

### 2.1 Climate simulations and selected forcing

Our configuration of CESM is a fully coupled general circulation model with atmosphere, ocean, sea ice, land, and runoff components (Hurrell et al., 2013). For the sake of computational feasibility, our climate forcing consists of time-slice simulations (similar to the Stone et al., 2013 methodology) at distinct points during the MIS-11 interglacial period and utilizes fixed modern-day ice sheets. Each time slice simulation utilizes $CO_2$, $CH_4$, and $N_2O$ levels characteristic of the selected time along with characteristic orbital parameters calculated based on the orbital solution by Laskar (2004). We assume static modern-day topography and land ice for all simulations, which are conducted at a spatial resolution of 2.5° longitude by 1.9° latitude for the atmosphere. Further details regarding the CESM time-slice simulations can be found in the methodology section of Crow et al. (2022). Bias corrections are calculated relative to climatologies from the final 100 years of a 400-year simulation under constant present-day (year 2000 CE) conditions.

Among the selected variables are monthly mean and standard deviation of 2-meter air temperatures converted to sea level, the atmospheric temperature downscaling lapse rate (described in greater detail in the following sub-section), the mean and standard deviations of zonal (U) and meridional (V) components of wind at a height relevant to orographic precipitation (details follow), the total precipitation, the total surface evaporation and sublimation, and ocean temperatures through approximately the top 600 m. GSM has a much higher spatial resolution than CESM and therefore captures more terrain variation, and its elevation profile is constantly recalculated in accordance with the dynamic ice sheet and lithospheric deformation.

Arrays of U and V winds were constructed utilizing data from various heights in the atmosphere, depending on the terrain profile. The goal was to capture wind direction and velocity at heights that are relevant for the generation of orographic precipitation. Since the majority of moisture transport occurs in the atmospheric boundary layer, our formula considers the wind interpolated to the CESM modeled surface height plus 500 m. Where this altitude lies below the height of the simulated 850 hPa pressure surface (i.e., low-altitude/coastal regions), 850 hPa winds are simply used. The input precipitation field is then adjusted by assuming a proportionality with the vertical velocity that such a wind field would induce, given the slope of

the terrain (Bahadory and Tarasov, 2018). This approximates the strong orographic forcing that steep slopes induce on precipitation and partly compensates for the mismatch in ice sheet model topography and the orographic boundary condition
used in the climate model.

Ocean temperatures are extracted at discrete levels through the top 600 m of the ocean column, roughly reflecting the present-day depth of waters along the continental shelf of Greenland. This depth also approximately corresponds to the depth of water that may have contact with marine-terminating outlet glaciers, thus exhibiting a strong influence on sub-shelf melt and calving.
The spatial resolution of CESM severely limits its ability to resolve fjord-scale ocean processes, so these temperatures represent only an approximation of the near-ice ocean environment.

All input fields are then linearly interpolated between the MIS-11 time slices. We acknowledge that this is an imperfect method that could fail to capture true peaks and nadirs of e.g. surface temperatures as they evolved through the MIS-11 interglacial,
as well as the possibility of abrupt and/or nonlinear climate transitions between the time slices. However, these time slices were chosen specifically to correspond to key points in the evolution of orbital forcing (precession minima and maxima, with strategically selected intermediate points), and the interpolation therefore should approximately capture the general evolution of climate through this period.

## 2.2 Temperature downscaling (lapse rate) methodologies

Since our climate simulations assume constant present-day ice and topography, there will be inherent contrasts between the land/ice surface heights in the climate and ice models. In order to address this discrepancy, a realistic vertical lapse rate must be utilized for correcting surface air temperatures to the appropriate elevation. We refer throughout this study to the surface slope-lapse rate (henceforth simply "lapse rate"), which is a lapse rate representing the rate at which surface temperatures vary at different surface altitudes. This is distinct from the free-air lapse rate, which represents the change in air temperature with
height through the atmosphere. It is thus more dependent on atmospheric dynamics and is often disconnected from surface characteristics.

As addressed previously, many modeling studies employ a fixed scalar lapse rate, such as the EISMINT3 standard of 7 K km$^{-1}$ (Huybrechts, 1997) or 6.5 K km$^{-1}$ (e.g., Viscaino et al., 2008). Piecewise lapse rates (Huybrechts and de Wolde, 1999) or
lapse rates as a tunable parameter (e.g., Stone et al., 2010) have also been used, but have no direct physical basis in modeled or observed temperatures. Therefore a prescribed lapse rate introduces a considerable source of error when coupling climate forcing to an ice model. Below, we describe the several different methods we tested in our study (in addition to a standard fixed lapse rate of 6.5 K km$^{-1}$).



### 2.2.1 Seasonally varying

The next logical step in complexity beyond a spatially and temporally uniform fixed lapse rate is a spatially-invariant, seasonally-varying lapse rate. Erokhina et al. (2017) utilized AOGCM simulations under preindustrial, early Holocene, and Last Glacial Maximum (LGM) conditions to demonstrate the dependence of the mean surface lapse rate over Greenland on not only the seasonal cycle, but also the large-scale climate forcing components (i.e., GHGs and orbital parameters). We adopt a similar methodology to Erokhina et al. (2017), utilizing least-squares regression of 2-meter climatological monthly air

temperatures from CESM against the CESM surface elevation, excluding points at elevations of less than 100 meters to eliminate contamination from oceanic grid cells. The slope of the regression line produced by each month's analysis then serves as the lapse rate that applies everywhere in our spatial domain for the given month.

### 2.2.2 Spatially and temporally varying (STV)

Our most sophisticated method is the fully spatially and temporally varying (STV) slope-lapse rate scheme, which is defined

on a point-by-point basis by examining the surface temperatures at all adjacent grid points. For each of the eight neighboring grid points (N, S, E, W, NW, SW, NE, and SE), the temperature delta is calculated and divided by the elevation delta. For elevation differences of less than 100 m, the lapse rate is set to 7 K km$^{-1}$, a representative mean slope-lapse rate value. This approach ensures that incidental temperature differences across a region with small elevation differences are not inordinately weighted compared to sites of more contrasting altitude. Points with effectively zero elevation difference (e.g., two adjacent

sea-level grid cells) are not considered in the calculation. For each grid point, the STV slope-lapse rate is the mean of all eligible surrounding slope-lapse rates.

In addition, we utilize a version of this method that is spatially smoothed with a radius of 3 CESM gridpoints (approx. ~300 km at 70°N latitude). The purpose of the smoothing is to reduce the effects of the poor representation of terrain along the

Greenland margin in CESM, minimizing any influence of abrupt gradients resulting from the exclusion of oceanic grid points and reducing the lapse rate gradient between different portions of the ice sheet.

### 2.2.3 Daytime-only STV

Finally, in an effort to account for diurnal cycle impacts, we calculated STV lapse rates based only on daytime 2 m air temperatures. Only a limited 5-year dataset of hourly values was available from each of the MIS-11c CESM simulations, and

no hourly data was available from the present-day simulation. The temperature bias corrections utilized for these simulations are therefore based upon the all-hours STV lapse rates. Hourly values corresponding to 6 am to 6 pm Greenland time were selected, approximately reflecting the window of maximum daily insolation. Differences in lapse rates, and therefore corrected sea-level air temperatures, are minimal during the darker and colder winter and spring months, but are substantial during



summer and early fall, when most ablation is occurring (Stone et al., 2010 similarly utilized a lapse rate based only on summer
temperatures).

## 2.3 GSM description

The Glacial Systems Model is a sophisticated thermomechanically coupled continental-scale ice sheet model that is designed
for large ensemble simulations of large ice sheets over glacial cycles (Tarasov et al., in prep.). It utilizes an evolved version of
the shallow-shelf/shallow-ice dynamical core (SSA/SIA) from Pollard and DeConto (2012) and Pollard et al. (2015).
Simulations herein were run at 0.5° longitude and 0.25° latitude grid resolution. Unique and/or noteworthy components of the
GSM include:

- a 4 km deep permafrost-resolving bed thermodynamics model that also corrects for seasonal snow cover of ice-free
  land areas (Tarasov and Peltier 2007);
- a global visco-elastic glacial isostatic adjustment (GIA) solver, updated from Tarasov and Peltier (1997);
- the orographic downscaling of precipitation using climatological wind fields (Bahadory and Tarasov 2018);
- and a novel inclusion of shortwave radiation fluxes into a traditional positive degree day (PDD) scheme.

The GSM has been utilized extensively in coupled ice-climate simulations, most commonly in a coupled system involving the
Earth-system model of intermediate complexity LOVECLIM (Goosse et al., 2010; Bahadory and Tarasov, 2018; Bahadory et
al., 2021). It has even been utilized previously to demonstrate the considerable spatial and temporal variability of near-surface
lapse rates over large ice sheets and the related dependence of ice volume evolution this causes (Bahadory and Tarasov, 2018).
It is therefore well-suited to the needs of this study.

Of further relevance is the recent completion of an approximate history matching (see Tarasov and Goldstein preprint in
EGUsphere 2023 for an explanation of history matching) of the last glacial cycle Greenland ice sheet with the GSM (Tarasov
et al., in prep.). This thereby provides a sample of GSM history-matched ensemble parameter vectors for which the non-climate
forcing components thereof can be used herein.

## 2.4 GSM parameters and boundary conditions

The parameters utilized by GSM to represent the various physical processes within and at the interfaces of the ice, till, and
bedrock of the domain are derived from an approximate history-matching routine. This glacial cycle history matching was
against deglacial and present-day observed data constraints of the GrIS. This set included relative sea level records, cosmogenic
age constraints, present-day ice thickness and horizontal surface velocities, deep ice core basal temperatures and the GRIP ice
core borehole temperature profile.

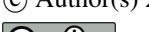



To partly address initialization uncertainties, the history matching simulations were run for two full glacial cycles (beginning around 240 ka). The history matching involved Markov Chain Monte Carlo sampling with Bayesian artificial neural network emulators along with over ten thousand full GSM simulations. A high-variance subset of history matched simulations provided not only the GSM parameter vectors but also the initialization state for the current simulations as described in the next section. Since our analysis uses a high-variance set of parameter vectors that were approximately history matched against deglacial and present-day observational constraints for the GrIS, a further examination of ice-model parameter sensitivity is not conducted here.

With the simulation domain being limited to Greenland and its immediate surroundings, a prescribed eustatic sea level was a required boundary condition. The LR04 sea level reconstruction (Lisiecki and Raymo, 2005) was employed for this purpose. Finally, GSM utilizes a handful of coupling parameters that modulate the degree to which temperature and precipitation inputs are bias-corrected. The values of these parameters were qualitatively tested to examine the effects of greater or lesser "blending" of input values, and ultimately were all set to utilize heavy bias correction.

## 2.5 Greenland ice sheet initialization

It has already been demonstrated that the ice volume derived from a given ice-climate simulation can be highly dependent on the initial ice topography and thermodynamic state (e.g. Rogozhina et al., 2011). We therefore opted to initialize our simulations from the previously described history-matched simulations of the past two full glacial cycles, which have e.g. bed thermal characteristics and bed deformation that are more representative than a steady-state integration from zero or from a present-day state. Our choice of selecting the 11.5 ka timeslice from the history matched simulations as the initial state for our MIS 11 simulations was inspired by Raymo and Mitrovica (2012; their figure 1), who presented an overlay of the evolution of the LR04 benthic stack $\delta^{18}O$ (Lisiecki and Raymo, 2005) from 440-410 ka and 30 ka-present. There is apparent similarity in the timing and magnitude of the transition from glacial to interglacial conditions at the start of MIS-11c and the present interglacial, and on this basis, 11.5 ka was chosen as an analogous point in the most recent glacial-interglacial transition to 423 ka, the start of our available forcing data.

The selected 10-member high-variance history-matched subset already contains a spread in ice volume, distribution, and thermal states at the 11.5 ka point. The initial sea-level equivalent (SLE) ice volumes at 11.5 ka in these simulations have a mean value of 11.1 meters, with a range of 9.2-12.2 meters. For reference, these 10 parameter vectors produce a mean GrIS SLE of 7.7 m at present day, compared to the estimated actual present-day water content of 7.4 m (Morlighem et al., 2017). The slight overestimation of GrIS volume at present-day is a common issue in ice sheet models and stems partly from the assumptions present in the shallow ice approximation physics (SIA; e.g., Stone et al., 2013, Stone et al., 2010) and partly from discretization.



Our initial states are integrated with constant 423 ka forcing for either 500 or 1500 years prior to the 423 ka begin date in order to avoid discontinuities from abrupt forcing changes in the period of interest. This "spinup time" is broadly similar to the approach taken by Mas e Braga and coauthors (2021) for their MIS-11c simulation of Antarctica. Each parameter vector is therefore represented twice in each ensemble, once with each spinup time. We therefore account for not only the inherent ice-state uncertainty in utilizing a variety of ice states associated with different ice model parameter sets, but additionally the uncertainty associated with selecting an analogue state from the present interglacial.

**2.6 Bias correction**

The version of CESM utilized in this study is understood to have a cold bias at high latitudes, at least in present-day and preindustrial climates (e.g., Wang et al., 2019). Given that unrealistically cold temperatures would be detrimental to accurately capturing the extent of GrIS surface melt in MIS-11c, an anomaly forcing approach was selected. For each time slice, the relative change of simulated temperatures and precipitation between the CESM present-day simulation and each MIS-11 time slice were calculated. Temperature anomalies are calculated as differences in sea-level-adjusted surface air temperatures, using the lapse rates calculated for each simulation in order to make the adjustment.

Precipitation bias correction is applied as a monthly-varying scale factor over Greenland and the surrounding continental shelf, representing the ratio between the modeled precipitation for a given time slice versus the present-day value. By default, only one scaling factor is applied over the entire ice sheet. However, the GSM provides the possibility to define sectors and calculate individual climatological scaling factors for each sector. We therefore tested the effects of using only one precipitation scaling factor against a two-sector north/south division that was established in an effort to address a consistent wet-bias pattern in southern Greenland. In general, the present-day CESM run is much wetter than both the reanalysis datasets around the perimeter of Greenland, and too dry in the center (Figure 1). However, south of 69°N, CESM has a large (25-75%) wet bias almost everywhere. This therefore serves as the dividing line in our two-sector precipitation tests.



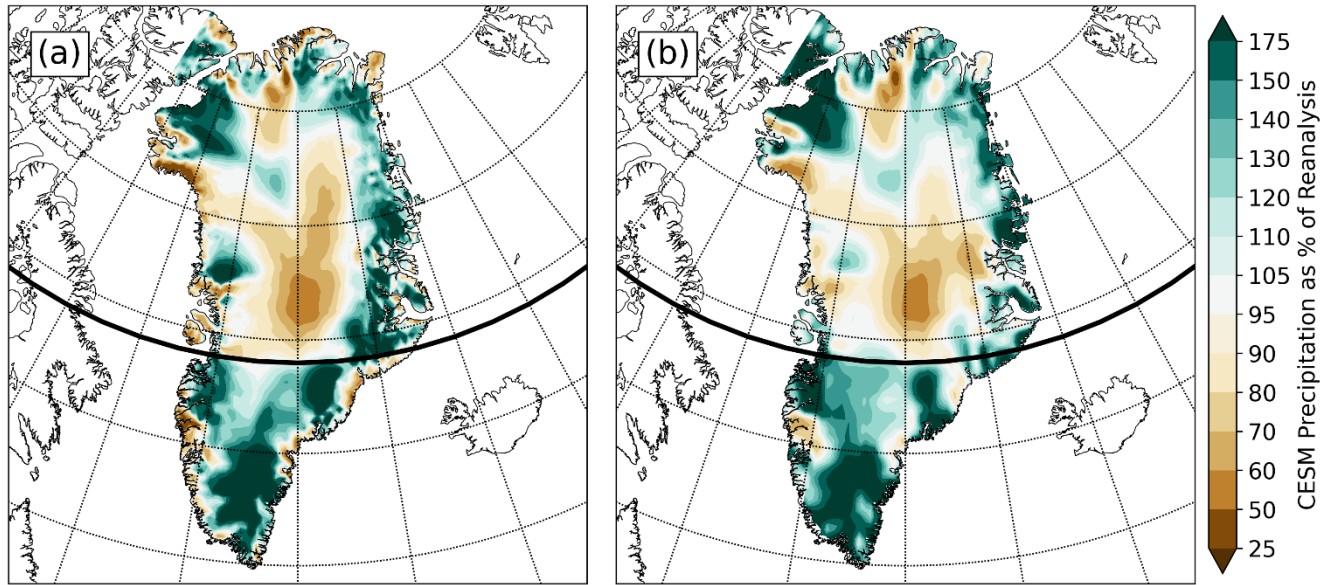


**Figure 1. Mean annual total precipitation bias ratio of the present-day CESM climatology to RACMO (a) and CESM to MAR (b). Green areas show where CESM is wetter compared to the regional reanalyses, brown where CESM is drier. The bold line of latitude is at 69°N, where the dividing line for 2-sector precipitation bias correction simulations was placed.**

Two different present-day regional climate model datasets were utilized as climatological baselines to which CESM anomaly forcing was applied: the Modèle Atmosphérique Régional v3.52 (MAR; Fettweis et al., 2017; Gallée and Schayes, 1994) and the Regional Atmospheric Climate Model v2.3p2 (RACMO; Noël et al., 2018). Both models have been developed specifically for use in polar regions and have been used extensively in ice sheet modeling studies previously (e.g., Carter et al., 2022, and references therein). Over Greenland, RACMO is slightly warmer (Figure 2) and drier (Figure 1) than MAR, leading to notable

differences in overall simulated ice volume.



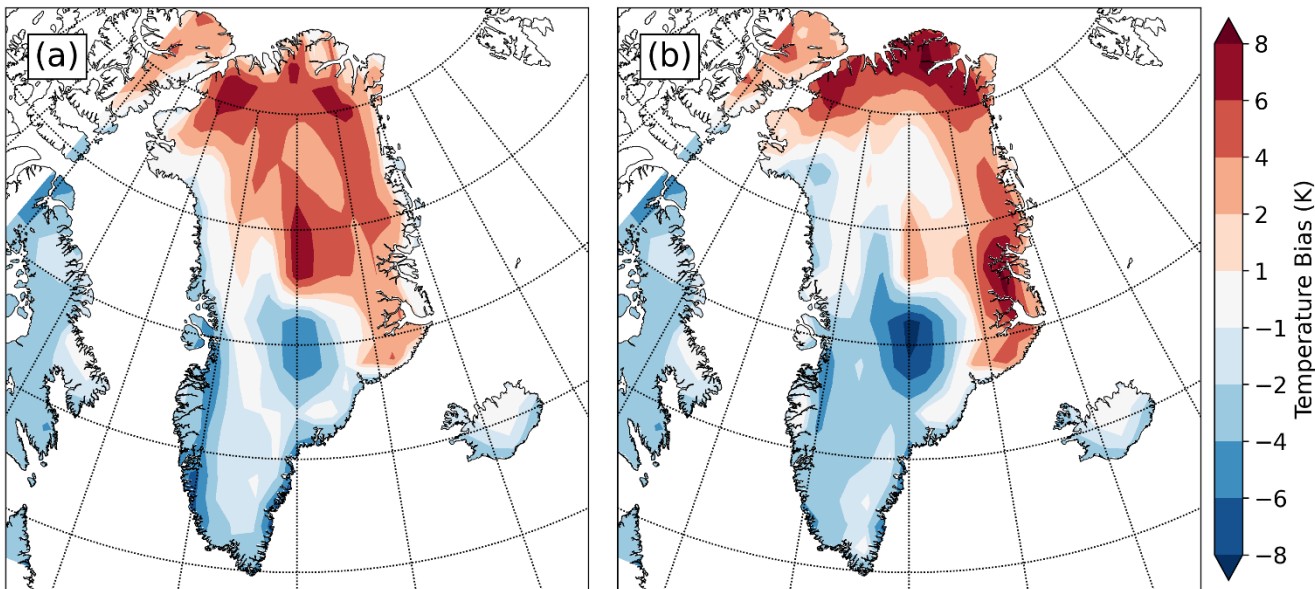

**Figure 2. Mean June-July-August sea-level converted temperature difference (bias) between RACMO present-day climatology and present-day CESM (a) and between MAR and CESM (b). Red colors indicate where reanalyses are warmer than CESM, blues indicate that the reanalyses are colder than CESM.**

## 3 Results

Numerous ensembles of simulations were conducted utilizing various combinations of the forcing methodologies described above. A summary table (Table 1) lists the present-day regional climate analyses to which the CESM anomalies were applied, the number of precipitation bias-correction sectors used, and the lapse rate method employed, as well as three key summary statistics. These are the fraction of ensemble members that preserve ice at the GSM grid cell corresponding to the Summit ice-core site throughout the entire simulation, the fraction of ensemble members that have zero ice depth at some point during the simulated period at the grid cell corresponding to the DYE-3 site, and the mean maximum SLE contribution from the melt of the simulated GrIS, averaged among all ensemble members.

| # | Temperature | Precipitation | PPT Sectors | Lapse rate method | Summit preservation | DYE-3 disappearance | Mean max SLE contribution |
|---|---|---|---|---|---|---|---|
| 1 | MAR | MAR | 2 | Seasonal | 20 / 20 | 4 / 20 | 3.27 m |
| 2 | RACMO | RACMO | 2 | Seasonal | 6 / 20 | 0 / 20 | 5.07 m |
| 3 | MAR | MAR | 1 | STV | 14 / 20 | 18 / 20 | 3.56 m |





| 4 | MAR | MAR | 2 | STV | 15 / 20 | 20 / 20 | 3.86 m |
|---|---|---|---|---|---|---|---|
| 5 | RACMO | RACMO | 1 | STV | 12 / 20 | 0 / 20 | 4.50 m |
| 6 | RACMO | RACMO | 2 | STV | 10 / 20 | 0 / 20 | 4.67 m |
| 7 | MAR | MAR | 2 | Smoothed | 6 / 20 | 4 / 20 | 5.60 m |
| 8 | RACMO | RACMO | 2 | Smoothed | 0 / 20 | 0 / 20 | 6.42 m |
| 9 | MAR | MAR | 2 | Daytime | 17 / 20 | 0 / 20 | 4.29 m |
| 10 | RACMO | RACMO | 2 | Daytime | 5 / 20 | 0 / 20 | 5.00 m |
| 11 | MAR | MAR | 2 | Constant | 20 / 20 | 0 / 20 | 2.20 m |

**Table 1. Summary of the selected forcing datasets and methodologies along with summary statistics for each ensemble. The "Summit preservation" and "DYE-3 disappearance" columns express the fraction of ensemble members (out of 20) that maintain >0 ice depth**
**at Summit throughout the entire simulation and members that achieve ice depth = 0 m at DYE-3 at some point during the simulation, respectively.**

The listed ensembles represent just a select subset of all the simulations that were conducted, which also included a number of sensitivity tests and cross-combinations of bias corrections (e.g., MAR temperatures with RACMO precipitation). The focus
is primarily on simulations featuring two precipitation sectors (north and south Greenland, divided at 69°N) because these were more successful at meeting our selection criteria, but for ensembles 3 and 5, included to illustrate the contrast with ensembles 4 and 6, respectively. Ensemble 11 utilizes a spatially and temporally uniform 6.5 K km$^{-1}$ lapse rate as a reference for the technique most commonly applied in other studies. The constant lapse rate simulations produce the least melt of the GrIS during MIS-11c and none of the 20 ensemble members meet both Summit and DYE-3 criteria.


Two trends are immediately apparent from the table: first, that for identical lapse rate methodologies, runs forced with anomalies from RACMO data generally produce a greater peak sea level contribution than those run with MAR (greater melting in MIS-11c associated with RACMO). This can be ascribed to the aforementioned slightly warmer and drier climatology in the RACMO dataset in comparison with MAR. Second, the dual "anchor point" criteria of Summit preservation
and DYE-3 disappearance prove difficult to simultaneously replicate, with only a minority of all simulations achieving both. This is not unlike the difficulties Yau et al. (2016) encountered in trying to simultaneously replicate temperatures at the NEEM and GISP2 core sites. Particularly problematic was achieving the complete melt of DYE-3, which retained ice in the overwhelming majority of all simulations outside of MAR-forced ensembles that utilized the STV lapse rates. This appears to be in part because of high accumulation rates across the South Dome region, causing it to maintain positive mass balance
throughout our simulated MIS-11c despite warmer-than-present temperatures.





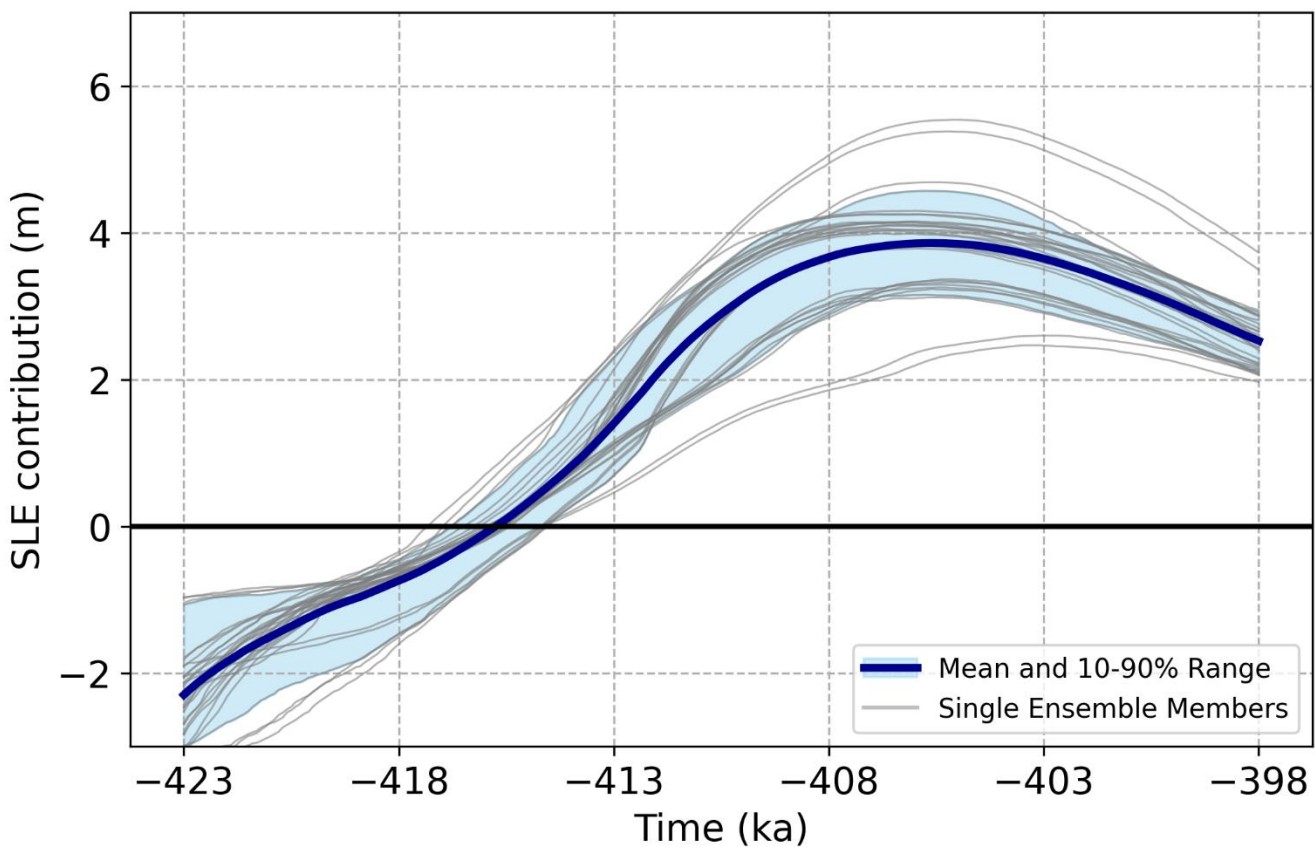

**Figure 3. Evolution of GrIS contribution to sea level relative to present (0 m line) for all 31 simulations that matched both Summit and DYE-3 conditions through the course of MIS-11c. The thick blue line represents the mean time-evolution of volume, while shading gives the 10th to 90th interquantile range. Light gray lines show each of the 31 individual member simulations comprising**
**the average.**

Across all the listed ensembles, a total of 31 simulations (14.1%) simultaneously met the Summit and DYE-3 criteria, all of which utilized MAR-based temperature and precipitation bias corrections and the large majority of which used STV lapse rates. These fitting simulations produce a mean peak GrIS SLE contribution of 3.9 m (10-90% range of 3.2-4.6 m) at a mean

time of 405.8 ka, the time-evolution of which is illustrated in Figure 3. This is equivalent to melting approximately 51% (range 41-60%) of the present-day Greenland ice sheet, based on the volume of the ice sheet from the present-day calibration simulations conducted with identical parameter vectors.

In order to better understand the differences incurred by each altered forcing factor, select ensembles are compared directly

below. The qualitative and statistical differences between them are discussed here in the context of each differentiating characteristic.





### 3.1 Initialization and spinup time differences

Assessed here are two forms of initialization uncertainty: different ice states stemming from the 11.5 ka realizations of the 10 selected GSM parameter vectors and the different relaxation times to account for the subjectivity of the 11.5 ka selection.

Differences in initialization state clearly have an effect, as simulations with a larger beginning ice volume tend to maintain larger ice volumes at their MIS-11c minima (not explicitly shown here, but partially recognizable among the individual members in Figure 3). The GSM parameters then appear to be the primary driver of the time-evolution of volume through the remainder of the simulations, as identical parameter vectors with different spinup times tend to follow nearly identical trends in time.


On the whole, our ensembles exhibit minimal sensitivity to the imposed differences in spinup time. Figure 4 illustrates the evolution of two 10-member ensembles utilizing the same 10 GSM parameter vectors and MAR bias corrections. After an initial difference in mean and spread at 423 ka arising purely from the use of a 500-year (blue) or 1500-year (red) spinup time (i.e., constant 423 ka forcing from the beginning of the simulation through 423 ka and starting from either 422.5 or 421.5 ka),

the two ensembles quickly converge. Only tiny differences between the ensembles can be observed after 419 ka. This pattern is robust to the choice of bias correction dataset (MAR or RACMO) and to various lapse rate methodologies, and even holds when examining only our criteria-matched simulations.




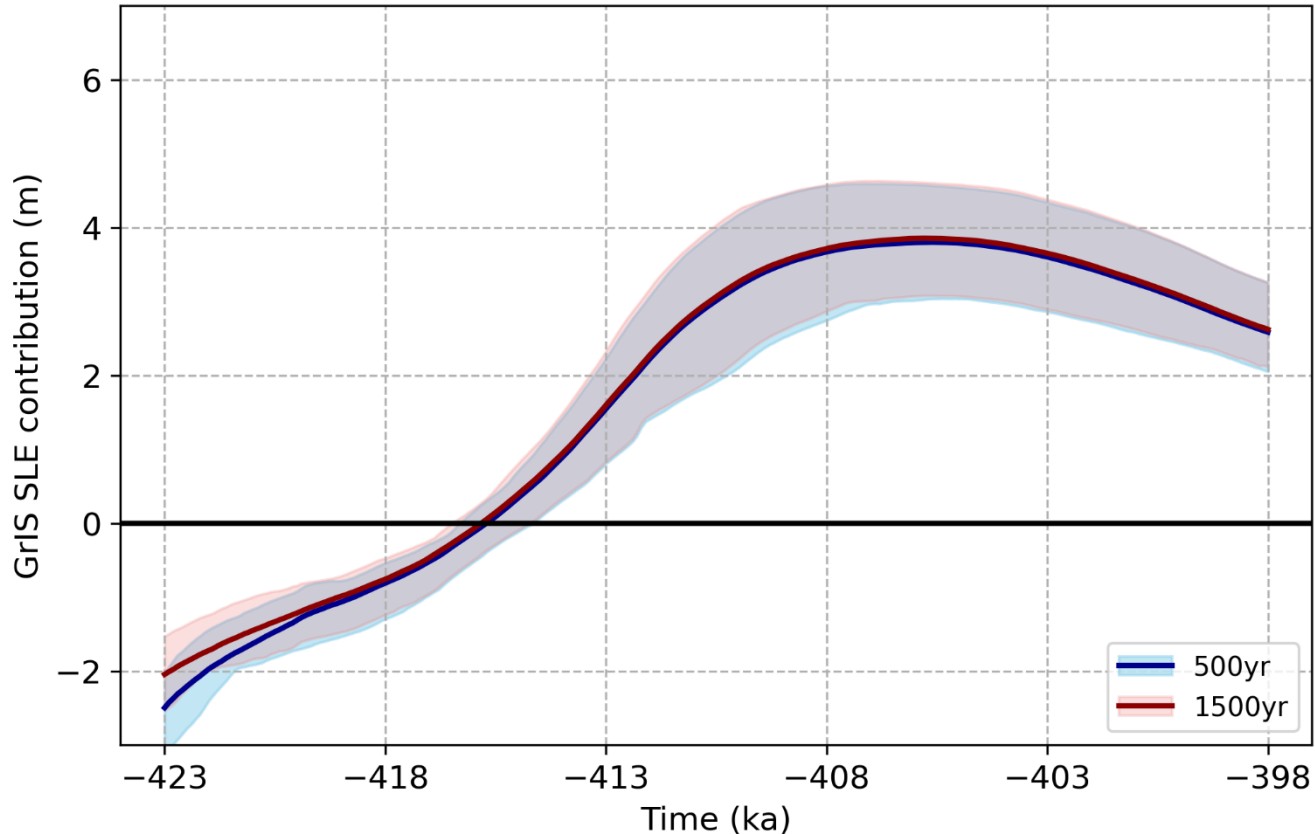

**Figure 4. Time evolution of sea-level equivalent contribution from Greenland in two ensembles run with identical parameter vectors**
**and forcing, but comparing the 500-year spinup time (blue) with the 1500-year (red). Ensemble means and spreads are practically indistinguishable after 418 ka.**

## 3.2 Climate forcing bias corrections

Unlike what we observe with comparing ensembles of different relaxation times, the differences between ensembles using either MAR or RACMO bias corrections are rather stark. Figures 5 and 6 demonstrate the time-evolution of the mean and
range of ensembles utilizing the same lapse rate techniques and precipitation bias correction sectors, but differentiating in their use of MAR (blue) or RACMO (red) bias corrections. As expected, the differences can be ascribed to the combined effects of the precipitation and surface temperatures on the surface mass balance.





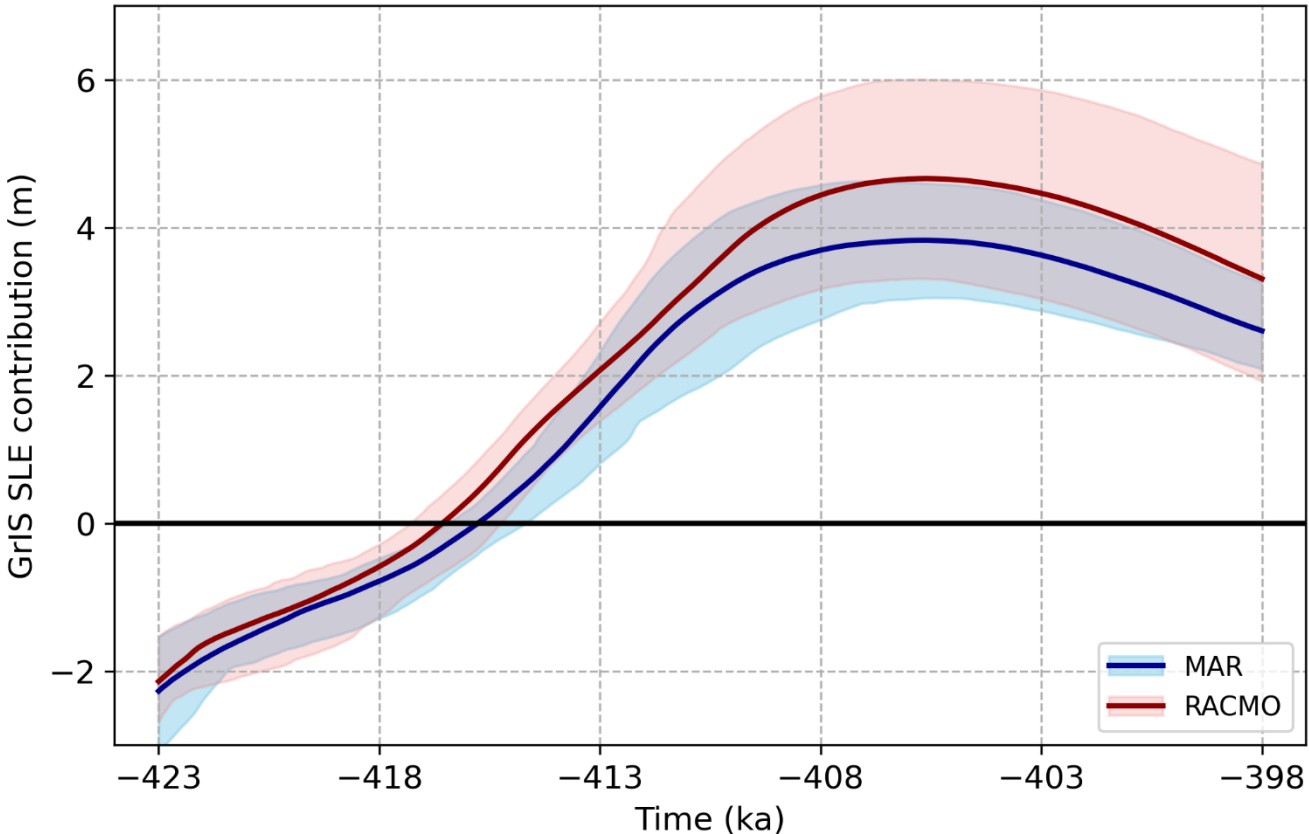

**Figure 5. Comparison of GrIS sea-level contributions from MAR (blue) and RACMO (red) ensembles utilizing fully varying lapse rates and 2 precipitation bias correction sectors. The shading represents the 10-90% range of each ensemble.**

A key result is that the melt extent associated with each forcing type also exhibits different sensitivities to different lapse rate techniques. The effects of the different lapse rates will be elaborated on in Section 3.4, but here we note the apparent amplification of contrasts between the MAR and RACMO ensembles using seasonally-varying lapse rates (Figure 6) as opposed to those using SVT (Figure 5). This sensitivity is a product of multiple factors, including the following:

- the MAR dataset over Greenland is slightly cooler and wetter than RACMO and the spatial patterns of each are slightly different;
- the seasonal cycles of the MAR and RACMO datasets are slightly different;
- the original MAR and RACMO datasets were of slightly different spatial resolution, thus raising the possibility of interpolation differences when both datasets are interpolated to the GSM grid. This could be particularly true along the steep marginal regions, which in turn exhibit the greatest influence on the size of the ablation zone.



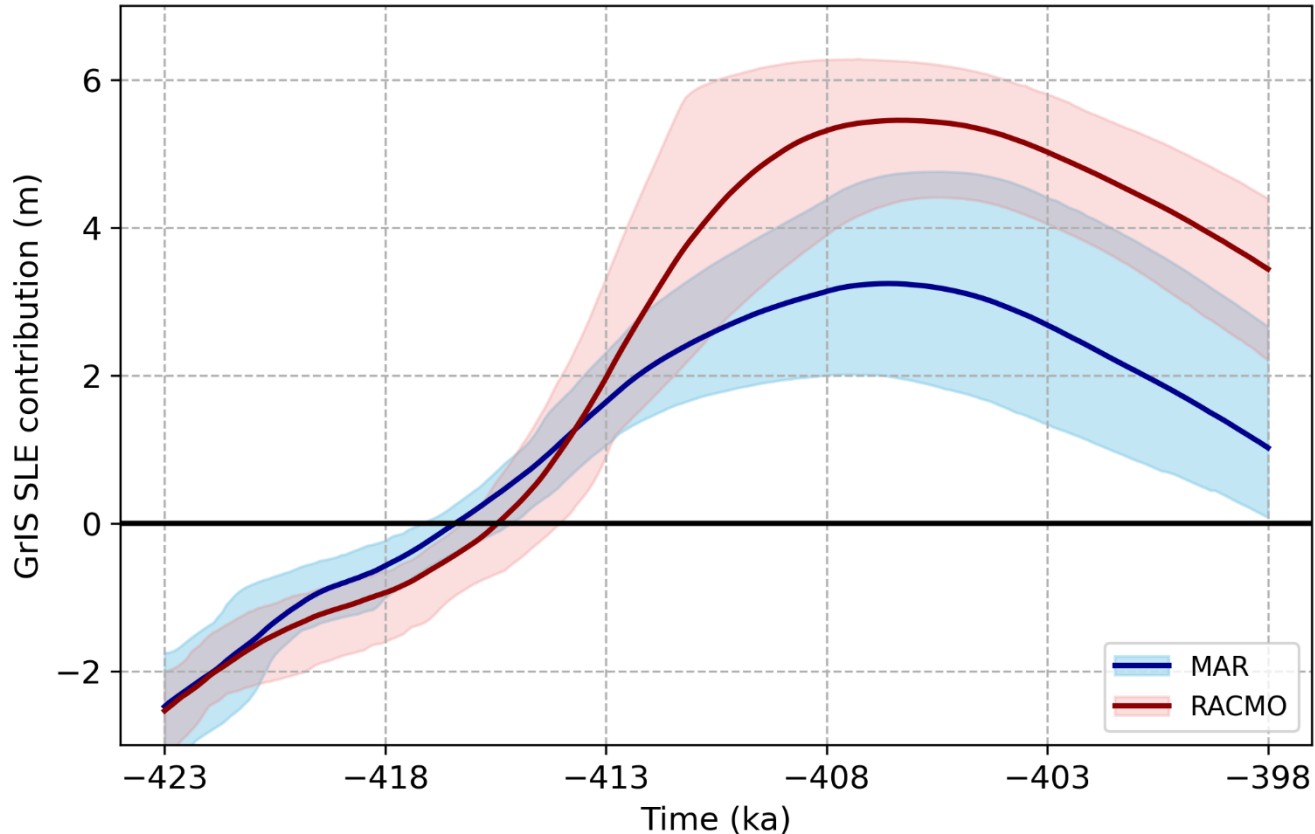

**Figure 6. As in Figure 5, but for seasonally-varying lapse rates.**

As an example, consider the two simulations depicted in Figure 7, which are selected from the ensembles depicted in Figure
      6. These runs utilize identical parameter vectors, initialization states, relaxation times, and the seasonally-varying lapse rate
      method. Stark contrasts exist in the ice states due to the bias correction differences, with the peak SLE contribution from the
      GrIS at around 5.5 m for the RACMO-corrected run and only 3.2 m for the MAR run. As expected, a substantial difference in
      the ablation zones is apparent; the ablation zone covers virtually the entire northern, western, and central portions of Greenland
by 405 ka in the RACMO run, whereas only northern marginal regions and the ice streams in the greater Jakobshavn Isbrae
      basin are net ablation zones in the MAR run. Looking at 415 ka surface ice velocities in the simulations offers insight into how
      this manifests in earlier stages of the ice evolution, with the RACMO run containing greater ice velocities and longer extensions
      of the ice streams into the interior regions of Greenland. The warmer surface temperatures in the RACMO analysis therefore
      contribute to a more thermodynamically imbalanced and deformable ice sheet in these simulations in comparison to those bias-
corrected with MAR data.



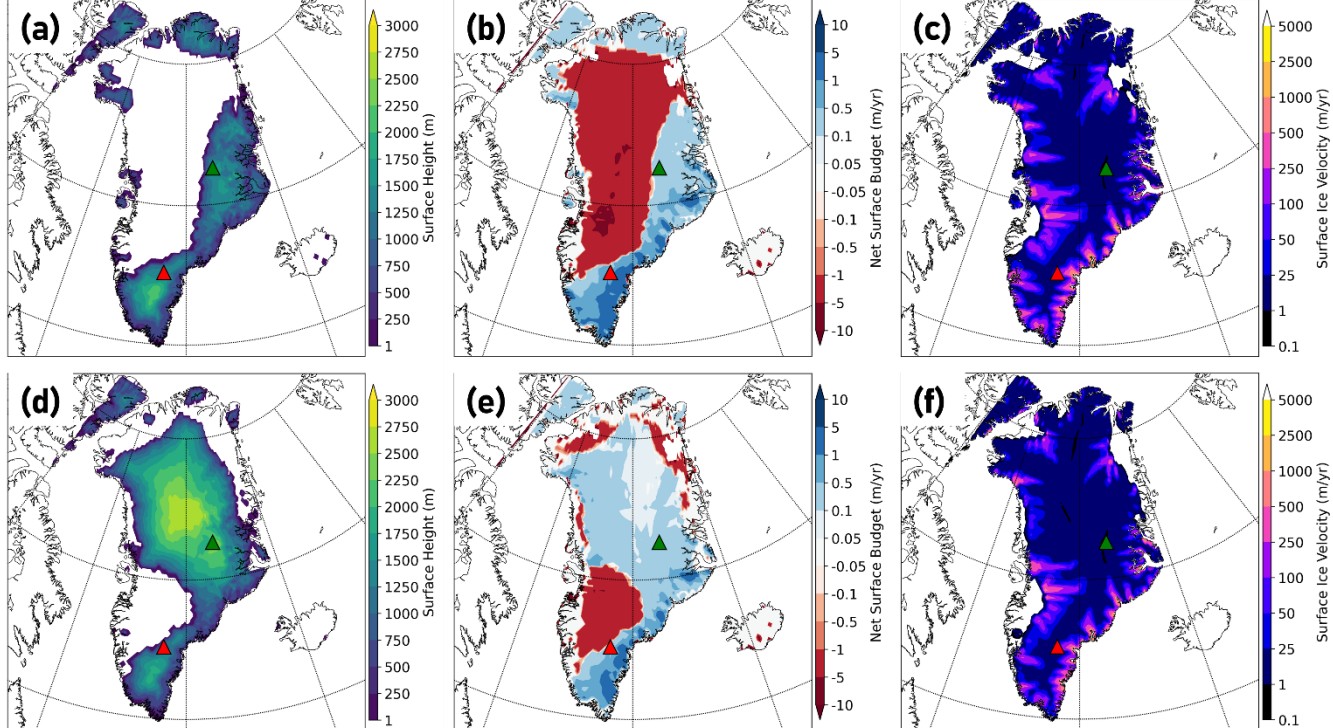

**Figure 7. Comparison of one model run from each of the ensembles depicted in Figure 6 with identical parameter vectors. Top row (a-c): temperatures and precipitation bias-corrected with RACMO data. Bottom row (d-f): bias corrections utilize MAR data. Left column (a and d): ice surface height in meters at year 405 ka of the simulation (near ice minimum). Center column (b and e): net surface budget at year 405 ka of the simulation, given in net meters per year of surface accumulation (positive, blue shading) or melt (negative, red shading). Right column (c and f): ice surface velocities expressed in meters per year. The locations of the Summit (green triangle) and DYE-3 (red triangle) core sites are depicted in each panel for reference.**

## 3.3 Precipitation scaling

The effects of using multiple precipitation bias-correction factors were also examined. As described in Section 2.6, the present-day CESM simulation is persistently wetter than both MAR and RACMO south of 69°N, so this division was utilized to enable the calculation of two separate monthly precipitation scaling factors. With all other variables kept constant (forcing type and lapse rate methodology), our simulations produce only minimal differences between the 1-sector and 2-sector forcing. Shown in Figure 8 is the difference between ensembles utilizing MAR forcing and STV lapse rates, with only a slight increase in mean melt contribution seen in the 2-sector simulations. Comparisons made between ensembles utilizing RACMO forcing are nearly identical, and thus not shown here.





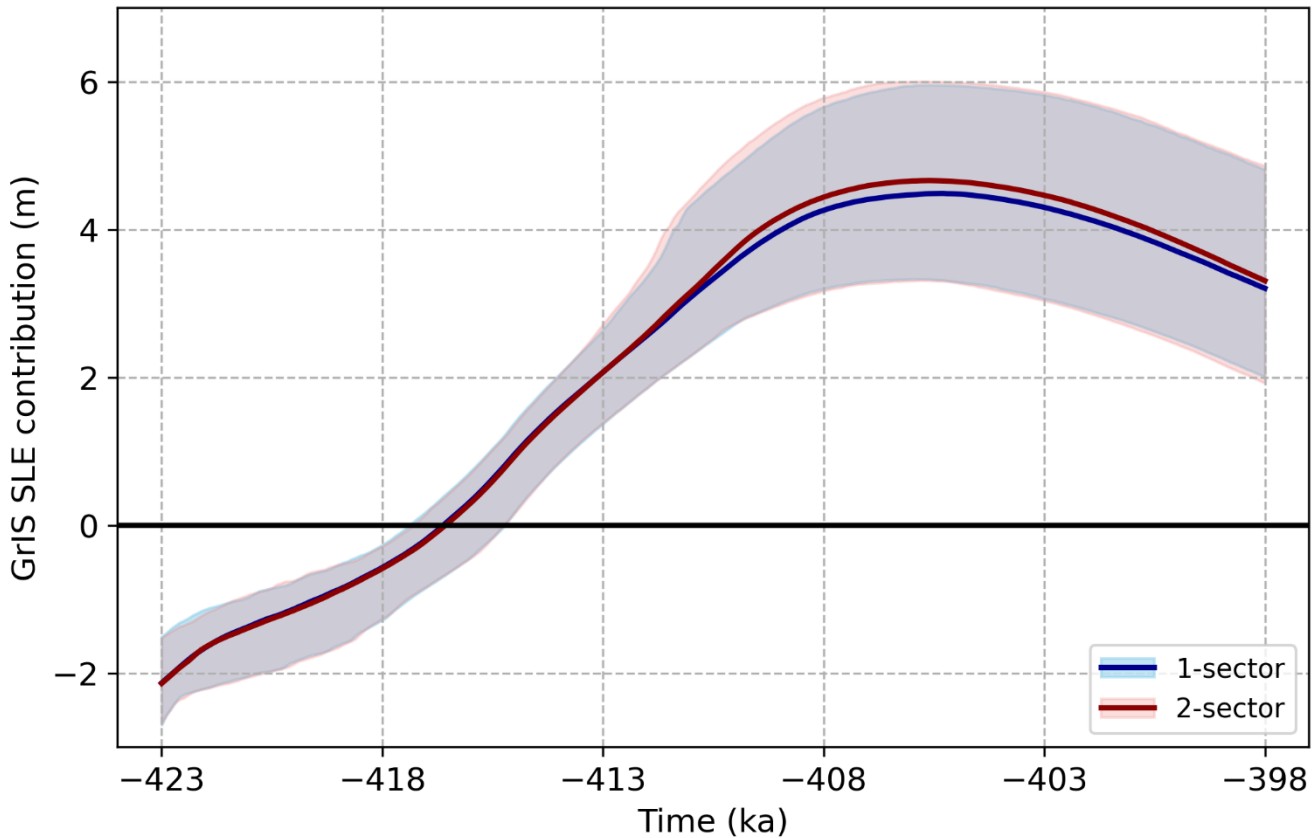

**Figure 8. Time evolution of sea level contribution from Greenland from two ensembles with MAR forcing and fully varying lapse rates, but using 1 precipitation scaling factor for the whole GrIS (blue) versus two sectors (red), divided at 69°N.**

This increase in sea level contribution is a result of slightly less positive mass balance from reduced precipitation over the South Dome region. The change also slightly improves our Summit and DYE-3 match rates, improving from 14/20 to 15/20 members preserving Summit and from 18/20 to 20/20 melting DYE-3 when utilizing MAR forcing and STV lapse rates (Table 1). While the differences between corresponding simulations in the two ensembles are difficult to see spatially, the small localized changes in surface mass balance ultimately result in improved representations of our key ice core locations.

**3.4 Lapse rate methodology comparison**

By far the most impactful forcing difference between simulations was the choice of lapse rate method. As illustrated in Figures 9 and 10, the resulting peak GrIS sea level contribution for each lapse rate method is distinctly different, ranging from approximately 2.2 m SLE with a constant 6.5 K km$^{-1}$ lapse rate to approximately 5.6 m SLE with the smoothed spatially and temporally varying method. The uncertainty ranges, characterized by the 10th to 90th percentile of individual members of each





ensemble, also vary in magnitude. The STV method provides the narrowest uncertainty range, while the smoothed and daytime
methods each span a range of over 3 m SLE around the time of minimum volume.

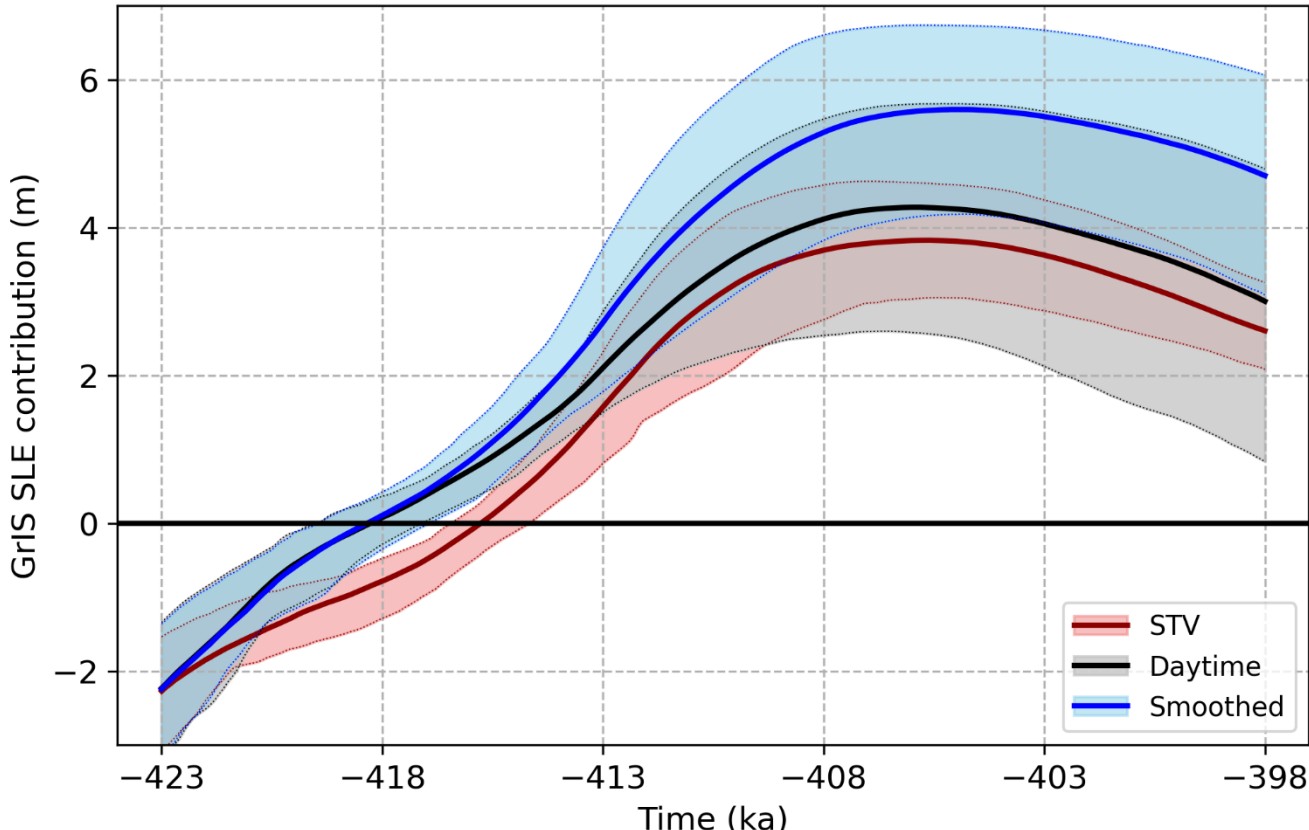

**Figure 9. Time-evolution of ensembles forced with MAR bias corrections, 2 precipitation sectors, and the STV (red), daytime-only (black/gray), and smoothed (blue) lapse rate methodologies. Shading represents the 10-90% range for each 20-member ensemble,**
**and bold lines give the time-mean evolution of all members in each ensemble.**

These differences can be primarily explained by the effect of the lapse rate on the surface mass balance. The fixed lapse rate method appears to result in an underestimation of the ablation zone and is not responsive to changes in orbital forcing, thus limiting the melt extent sharply. The seasonal lapse rate is spatially invariant and therefore has nearly the same limitation as
using a fully fixed lapse rate: the higher vertical temperature gradients in coastal/marginal zones are not resolved, reducing the extent of the ablation area. Higher gradients occur in these marginal zones during the warm season primarily due to the albedo differences between low-elevation, snow- and ice-free regions and the snow- and ice-covered higher elevations.



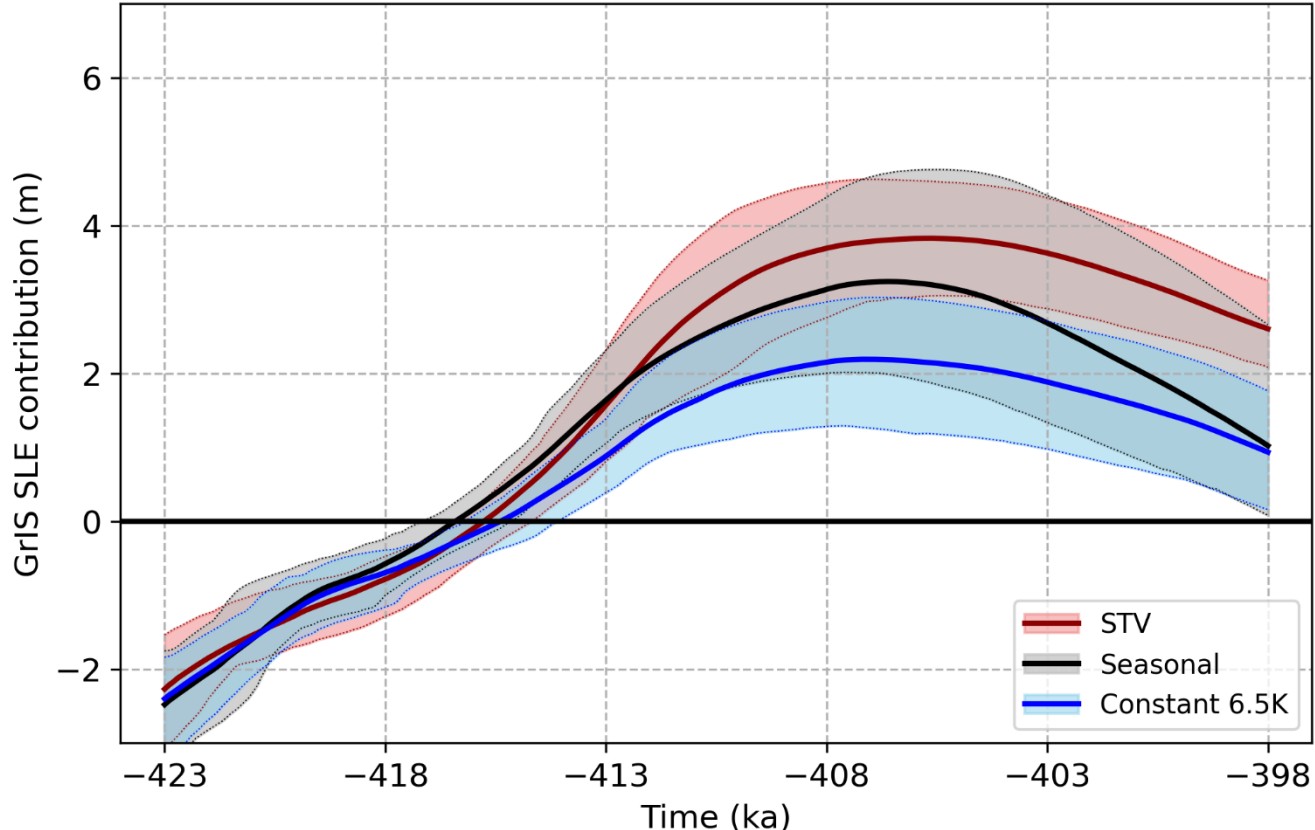

**Figure 10. As in Figure 9, but for comparison of the STV, seasonally-varying, and constant 6.5 K km$^{-1}$ lapse rate methods.**


On the other end of the spectrum, the smoothed lapse rates result in a likely overestimated ablation zone, as the large lapse rates that generally occur around the steep margins are artificially broadened into more central portions of the ice sheet. It is also unsurprising that the daytime lapse rates result in greater melt than the standard STV method, given that daytime-only lapse rates will tend to be higher than all-hours averages. Daytime lapse rates are higher than equivalent all-hours lapse rates

due again to albedo differences between higher and lower altitudes, plus the removal of the overnight hours in which the surface energy budget is dominated by longwave cooling.

However, higher lapse rates do not automatically translate into greater ablation, as increased lapse rates result in cooler simulated conditions over higher terrain, helping to preserve high-altitude and central portions of the ice sheet. The interplay

between the high, cold interior region and the low, warm marginal region explains much of the very large spread in volumes across individual members of the daytime and smoothed lapse rate ensembles. Small differences in initial ice extent and elevation are amplified throughout the simulations, resulting in the large spread seen in both the daytime and smoothed ensembles.





The STV lapse rate methodology produces the narrowest spread amongst ensemble members. The spatial pattern of the calculated lapse rate is broadly similar for each MIS-11c time slice, reflecting the constancy of the ice margins throughout the fixed-ice simulations, but with small variations dependent on the orbital forcing changes. This method is the most physically justifiable, as it accounts for regional patterns of temperature and lapse rate driven not only by terrain differences, but also differences in regional climate regimes. Consider, for example, the very stormy and subarctic southern reaches of Greenland

and the very dry Arctic northern slope of Greenland, which is often characterized by shallow polar high pressure and even temperature inversions. Accounting for the presence of such features enables a more physically consistent coupling of temperatures between CESM and the dynamic ice of GSM.

## 4 Discussion

    In this study, we have presented an examination of the relative impacts of various sources of uncertainty in coupled ice-climate

modeling with specific application to MIS-11c. Though uncertainties associated with numerically approximated ice processes remain and are not directly addressed here (e.g., Goelzer et al., 2017), we have demonstrated that climate forcing and its downscaling is overwhelmingly the dominant influence on our simulated GrIS. Ideally future studies should strive to use fully two-way coupled climate-ice sheet simulations, thus reducing or eliminating many of these uncertainties. For the time being, however, large ensembles remain a useful tool for uncertainty assessment, and for computational practicality reasons this

remains the domain of EMICs and one-way coupled simulations.

    Our use of a full AOGCM for climate forcing offers the benefit of sophisticated, relatively high-resolution climate forcing, but at the expense of non-interactive, prescribed ice. This in turn means that surface albedo and vegetation feedbacks are missing from our climate forcing, thus impacting the temperature forcing (and to a lesser extent precipitation). The effect of the missing

feedbacks on temperature, for example, could be to underestimate the lapse-rate feedback effect (Pritchard et al., 2008). As ice retreats, particularly in marginal and low-elevation zones, the surface albedo will tend to increase, and the emergence of silty layers and eventually bedrock will result in an altogether radiatively different surface. Additionally, the time-slice methodology allows for the possibility of missing peak climate forcing conditions, which, while unlikely, could have potentially occurred between chosen time slices. While our time slices were strategically selected to reflect precession minima,

maxima, and intermediate points in the precession cycle, the fact that our forcing is not continuous allows for possible underestimation of peak interglacial warmth. The MIS-11c simulations presented here are therefore likely conservative, and skew towards the lower bound of possible GrIS melt for this period.

    Achieving both filtering criteria simultaneously with our simulations proved difficult, as generally high accumulation rates

around DYE-3 often prevented complete melt and high ablation rates at Summit often resulted in elimination of ice there. That



our simulations that produced the greatest overall melt of the GrIS in MIS-11 (those utilizing RACMO bias corrections and smoothed lapse rates) had zero success at meeting either criteria illustrates the highly uncertain retreat pattern of the GrIS. By utilizing two separate precipitation bias corrections, one applied to Summit and the other applied to DYE-3, we were able to achieve a modest improvement in meeting these criteria. To the authors' knowledge, no other studies have used such a variable
bias correction in their investigations, but it clearly has utility when there are significant regional or sub-regional model biases, which is generally the case for all past and current climate models.

Furthermore, differences in initialization states had only minor impacts on the exact spatial ice distribution of the simulated GrIS, but ultimately little impact on its estimated sea level contribution during MIS-11c. A spinup time lead-in of 500 or 1500
years prior to the beginning of our climate forcing period was also of very minor significance, with runs utilizing identical parameter vectors quickly converging after a few thousand years. This may be contrary to reader expectations, given that several previous studies (e.g., Rogozhina et al., 2011; Aschwanden et al., 2013) have identified initialization states as a key factor in modeled ice sheet outcomes. However, the short (500 year) versus long (1500 year) spinup times represent primarily the uncertainty due to the uncertain choice of simulation start time from the present-day spinup simulations. The other aspects
of initialization uncertainty, e.g., differences in initial ice distribution and temperature, are inherently accounted for in the use of multiple parameter vectors and their corresponding initial states. The initial spread between different parameter vectors dominates any minor effects from the short versus long spinup times.

We have also demonstrated that proper coupling and bias-correction of near-surface temperatures is of paramount importance
to simulating the paleo-GrIS, as it exerts a critical control on surface mass balance. Bias corrections (against higher-quality or higher-resolution datasets) are an optional but very useful means of helping to constrain the uncertainties introduced by utilizing climate forcing from models with known temperature biases or other deficiencies (e.g., Fyke et al., 2011; Ridley et al., 2010). If opting for a bias-corrected or anomaly-forcing method, then selecting baseline datasets that are optimized for polar climates is also advisable (Carter et al., 2022).


Ultimately, nothing exhibited such a great influence over our GrIS simulations as the chosen slope-lapse rate technique, and for two primary reasons: (1) the dual manifestation of the lapse rate in both the actual temperature forcing applied and the bias corrections and (2) the overwhelming influence of temperatures upon the surface mass budget. The calculated lapse rates influence temperatures twice: first, in the correction of the surface temperatures from CESM to the appropriate ice-surface
height as calculated by GSM; and second, in the magnitude of the applied temperature bias correction, as the MAR and RACMO temperatures were themselves converted to sea-level temperature for direct comparison with CESM. Furthermore, temperature downscaling methodology need not be a subjective and arbitrary choice of scalar lapse rate value; rather, we have demonstrated here that data-based and observationally-supported alternatives are readily available. We have found a data-based spatially and temporally varying lapse rate to be the optimal solution.



## 5 Conclusions and outlook

This study was conducted with the dual goals of (1) offering additional constraints on the Greenland ice sheet's contribution to sea level rise during the MIS-11c interglacial and (2) addressing the previously under-examined influence of bias correction and coupling of climate forcing on simulated ice sheets. In particular, we have emphasized the impact of the choice of methodology by which surface temperatures are downscaled from the climate model to the dynamic ice surface in the ice model, demonstrating that it has a dominant effect on the simulated ice sheet.

To the first point, we have found that the minimum volume of the GrIS during MIS-11c was likely slightly less than half of its present-day value. Our simulations matching the criteria of (1) Summit preservation and (2) DYE-3 melt resulted in a mean maximum contribution to the MIS-11c sea level highstand of 3.9 m from the Greenland ice sheet, peaking around 405.8 kya. The uncertainty range defined by the middle 80% of matched simulations is a SLE contribution of 3.2-4.6 m. This estimate, which is likely somewhat conservative, is on the low side of existing estimates. Qualitative estimates based on paleodata have suggested a GrIS contribution of 4.5 to 6.0 m of sea level rise in MIS-11c (Reyes et al., 2014).

Somewhat more comparable is the modeling study of Robinson et al. (2017), which utilized the same two constraining criteria for filtering simulations (preservation of ice at Summit and complete melt at DYE-3). However, their REMBO climate model is a vertically-integrated energy balance model (Robinson et al., 2010) and therefore lacks any atmospheric dynamics. Furthermore, they use a scalar 6.5 K km$^{-1}$ lapse rate for temperature-elevation corrections (Robinson et al., 2010). Their ensemble simulations produced a contribution estimate of 3.9 to 7.0 m of sea-level rise, but without accounting for the potential uncertainties introduced by the scalar lapse rate nor those from the highly simplified climate model. Some of the discrepancy can likely be explained by the fact that their simulations produce a greater duration of melt conditions, with the peak mean SLE contribution occurring approximately 3 kyr later than ours (402.8 ka). While the overall temperature anomalies around Greenland are similar between the studies (Figure 1d in Crow et al., 2022), the temperatures interpolated between CESM runs appear to be somewhat lower than those from REMBO in the Robinson study in the 408-398 ka period, thus enabling more late-interglacial melt in the latter.

The strong dependence of the Greenland ice sheet produced by each simulation on the chosen lapse rate methodology for vertical downscaling of 2 m air temperature highlights the importance of this often-neglected source of uncertainty in coupled ice-climate modeling. Our simulations utilizing the common but observationally and physically unjustifiable choice of a scalar lapse rate (in our case 6.5 K km$^{-1}$) produce the least melt of any of the temperature downscaling methodologies presented here. Constant lapse rates, and even seasonally-varying but spatially uniform lapse rates, fail to capture critical differences in regional climate conditions and therefore underestimate the extent of marginal ablation regions. In contrast, a spatially- and temporally-varying lapse rate, calculated from the climate model temperature and elevation data, can capture the seasonal



cycle and regional climate differences in a physically realistic (albeit subject to model biases) way. Further improvements to the scheme presented here could be made by utilizing climate simulations with fully interactive ice sheets (whether via online
or offline asynchronous coupling). Future modeling studies should strongly consider implementation of similar coupling methodologies in order to avoid further compounding errors inherent to climate models.

**Data availability.** Data and analysis code can be obtained by contacting the corresponding author, and an upload of related data is planned for later in 2023.


**Author contributions.** This research was performed as part of the PhD studies of lead author Brian Crow, who is advised by authors Dr. Matthias Prange, Prof. Dr. Michael Schulz, and Prof. Dr. Lev Tarasov. BC carried out climate model simulations with advice and code contributions from MP and experimental design input from MP and MS. The experimental design for ice sheet model simulations was developed by LT and BC. Ice sheet model simulations were carried out by LT. All analysis
and figure creation were conducted by BC.

**Competing interests.** Authors L. Tarasov and M. Prange are members of the editorial board of the Icy Landscapes of the Past special joint issue of Climate of the Past, The Cryosphere, Earth System Science Data, and Earth Surface Dynamics. The peer-review process was guided by an independent editor, and the authors have no other competing interests to disclose.


**Acknowledgements.** The authors would like to thank the Deutsche Forschungsgemeinschaft (DFG) and the Natural Sciences and Engineering Research Council of Canada (NSERC) for furnishing funding for this research via the ArcTrain International Research Training Group. For providing climatological outputs from the present-day CESM run referenced in this study, we wish to express our gratitude to Ute Merkel of the University of Bremen. We are grateful to both the Northern German
Supercomputing Alliance (HLRN) and Digital Research Alliance of Canada and ACENET for access to their computing resources for the climate and ice simulations in this study, respectively. This study also contributes to the DFG Cluster of Excellence EXC-2077 – 390741603 and the PalMod German Climate Modeling Initiative.

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
