# Peer review of "Uncertainties originating from GCM downscaling and bias correction with application to the MIS-11c Greenland Ice Sheet"

_EGUsphere, 2023_

## Author Response (AR1)

Reviewer 1 Response Comment

The authors would like to thank the anonymous reviewer for their thoughtful and positive response to our manuscript. We are pleased to see that the primary messages of our manuscript were well-received. Most suggestions will be fully implemented. The specific points of constructive criticism have been addressed as follows:

- L80-85: Thank you for this reference, and we are glad to see that its results are consistent with ours. Mention will be added to the manuscript.
- L117: Specifics of the time slices have now been added to this section.
- L118: Reference for the GHG concentrations has been added.
- L125: This sentence has been edited for clarity.
- L259: Yes, and the relevant sentence will be edited to include mention of this fact.
- L270: This sentence will be edited and expanded for clarity.
- L276: Methodology will be more clearly described here.
- L338: Percentages will now be included in this section.
- L426: Comparing figures 5 and 6 illustrates that the magnitude of the difference in sea level contribution between the MAR and RACMO datasets varies widely depending on the lapse rate technique utilized. Therefore the choice of lapse rate technique is the first-order criteria affecting the melt magnitude and the dataset choice is secondary.
- L436: A figure showing the four lapse rate types will now be included. We have chosen to show the 413 ka timestep, however, as this is during the height of the GrIS melting phase, and the contrasts are more clearly illustrated.
- Fig. 2: Colors have been reversed, as requested.
- Table 1: Columns have been combined.

Reviewer 2 Response Comment

The authors would like to thank this anonymous reviewer for their very positive assessment of our manuscript and their recommendations for improvement. We are pleased to see that our core messages were well understood. The recommended changes are few and will therefore be addressed point-by-point below.

- L19: Respectfully, this is not correct. We follow long-established field precedent in using "methodology" to mean "a collection of methods or techniques used in a study or field of study," a definition which is supported by various English dictionaries (e.g., Cambridge: https://dictionary.cambridge.org/dictionary/english/methodology).
- Introduction: Thank you to the reviewer for the recommended reference. A brief discussion of other coupling strategies, such as dynamical downscaling, will now be included.
- L37: The 6-13 m in the abstract refers to the overall MIS-11 sea level highstand. The reference on line 37 addresses the Greenland-only sea level contribution during MIS-11. This sentence will be lightly edited for clarity.
- L183: The fixed 7 K lapse rate for areas with small elevation deltas applies mostly to grid cells that are at least fractionally oceanic and therefore is of minimal consequence for surface temperatures of the ice sheet. Other lapse rates were therefore not tested.
- L270: The GSM already contains a physically-motivated orographic correction to precipitation inputs. The surface elevation and winds a few hundred meters above the surface are used to diagnose vertical velocities that lead fairly directly to the local orographic scaling factor with subsequent scaling to ensure regional mass-conservation. While this does not completely overcome the biases of the input data (i.e., CESM), a further orographic bias adjustment would be redundant. While the employed bias correction scheme is simplistic, the highly nonlinear nature of the dynamics responsible for precipitation do not lend themselves well to scalar adjustments. Discussion of this point has been expanded in section 2.6.